# Unraveling the Molecular Basis of Color Variation in *Dioscorea alata* Tubers: Integrated Transcriptome and Metabolomics Analysis

**DOI:** 10.3390/ijms25042057

**Published:** 2024-02-08

**Authors:** Yue Wang, Rui-Sen Lu, Ming-Han Li, Xin-Yu Lu, Xiao-Qin Sun, Yan-Mei Zhang

**Affiliations:** 1Institute of Botany, Jiangsu Province and Chinese Academy of Sciences, Nanjing 210014, China; yuewang@cnbg.net (Y.W.); lurs@cnbg.net (R.-S.L.);; 2Jiangsu Key Laboratory for the Research and Utilization of Plant Resources, Nanjing 210014, China

**Keywords:** *Dioscorea alata*, tuber color, transcriptome, metabolomics, flavonoids, antioxidant activity

## Abstract

*Dioscorea alata* L. (Dioscoreaceae) is a widely cultivated tuber crop with variations in tuber color, offering potential value as health-promoting foods. This study focused on the comparison of *D. alata* tubers possessing two distinct colors, white and purple, to explore the underlying mechanisms of color variation. Flavonoids, a group of polyphenols known to influence plant color and exhibit antioxidant properties, were of particular interest. The total phenol and total flavonoid analyses revealed that purple tubers (PTs) have a significantly higher content of these metabolites than white tubers (WTs) and a higher antioxidant activity than WTs, suggesting potential health benefits of PT *D. alata*. The transcriptome analysis identified 108 differentially expressed genes associated with the flavonoid synthesis pathway, with 57 genes up-regulated in PTs, including *CHS*, *CHI*, *DFR*, *FLS*, *F3H*, *F3′5′H*, *LAR*, *ANS*, and *ANR*. The metabolomics analysis demonstrated that 424 metabolites, including 104 flavonoids and 8 tannins, accumulated differentially in PTs and WTs. Notably, five of the top ten up-regulated metabolites were flavonoids, including 6-hydroxykaempferol-7-*O*-glucoside, pinocembrin-7-*O*-(6″-*O*-malonyl)glucoside, 6-hydroxykaempferol-3,7,6-*O*-triglycoside, 6-hydroxykaempferol-7-*O*-triglycoside, and cyanidin-3-*O*-(6″-*O*-feruloyl)sophoroside-5-*O*-glucoside, with the latter being a precursor to anthocyanin synthesis. Integrating transcriptome and metabolomics data revealed that the 57 genes regulated 20 metabolites within the flavonoid synthesis pathway, potentially influencing the tubers’ color variation. The high polyphenol content and antioxidant activity of PTs indicate their suitability as nutritious and health-promoting food sources. Taken together, the findings of this study provide insights into the molecular basis of tuber color variation in *D. alata* and underscore the potential applications of purple tubers in the food industry and human health promotion. The findings contribute to the understanding of flavonoid biosynthesis and pigment accumulation in *D. alata* tubers, opening avenues for future research on enhancing the nutritional quality of *D. alata* cultivars.

## 1. Introduction

*Dioscorea alata* L., also known as the water yam, is an edible shallow root tuber belonging to the Dioscoreaceae family and is valued for its dietary carbohydrates, amino acids, and essential minerals [1]. It is widely cultivated in tropical and subtropical regions and is one of the most economically important crops, serving as a staple food for millions of people [1,2,3]. The tuber flesh of *D. alata* exhibits a range of colors, varying from white, yellowish, orange, and pink to purple [4]. The diversity of nutrients and bioactive compounds quantitatively varies within the same species [5,6]. Recent studies have highlighted the importance of the tuber flesh color as a key quality trait of yams, with color variation generally attributed to differential metabolic profiles [7,8]. Among the pigments responsible for the plants’ coloration, flavonoids play a prominent role [9,10,11,12]. Flavonoids possess antioxidant and anti-inflammatory properties, and their abundance in food products can enhance both the nutritional value and appeal to consumers, owing to the flavonoids’ associated health benefits [13,14]. Among them, anthocyanins are perhaps the best characterized flavonoids, with their important role in plant physiology, and anthocyanins are the most abundant flavonoid compounds and are responsible for the vibrant colors of flowers, fruits, vegetables, and even cereals [10,11,12,15]. Therefore, understanding the differences in flavonoid biosynthesis, especially anthocyanins and their precursors among *D. alata* varieties exhibiting different colors, is essential for optimizing the nutritional value and health-promoting attributes of *D. alata*.

The biosynthesis of flavonoid compounds occurs downstream of the phenylpropane pathway in plants. According to the degree of oxidation of the heterocyclic ring and the number of hydroxyl or methyl groups on the benzene ring, flavonoids can be classified into 12 subgroups, including chalcones, flavanones, flavones, isoflavones, flavonols, dihydroflavonols, leucoanthocyanidins, and anthocyanins [16,17]. In the past decade, genes that encode important enzymes and transcription factors that are responsible for flavonoid biosynthesis have been cloned from different plants [18,19,20,21]. The main transcriptional regulator in flavonoid biosynthesis is the MBW complex, comprising myeloblastosis (MYB), the basic helix–loop–helix (bHLH), and the WD-repeat protein (WD40). The MBW complex regulates the accumulation of flavonoids by directly targeting and activating the expression of flavonoid synthesizing genes, such as *chalcone synthetase* (*CHS*), *chalcone isomerase* (*CHI*), *flavono-3-hydroxylase* (*F3′H*), *dihydroflavonol-4-reductase* (*DFR*), and *anthocyanin synthetase* (*ANS*) [22,23,24,25]. Dihydroflavonol serves as a crucial branch point in the flavonoid biosynthesis pathway, acting as the common precursor for flavonols, anthocyanins, and proanthocyanidins. F3H catalyzes the conversion of naringenin, eriodictyol, and pentahydroxyflavanone into their corresponding dihydroflavonols, including dihydroflavonols, dihydrokaempferol (DHK), dihydroquercetin (DHQ), and dihydromyricetin (DHM) [26,27]. DFR is a key enzyme that catalyzes the reduction of DHK, DHQ, and DHM to produce different leucoanthocyanidins, including leucopelargonidin, leucocyanidin, and leucodelphinidin, respectively [28]. Subsequently, under the catalysis of ANS, colorless leucoanthocyanidins are transformed to corresponding anthocyanidins [27]. The promotion or suppression of any one of the enzymes catalyzing a series of reactions that make up a pathway will change its final product [1]. Although the biosynthesis and regulatory mechanisms of flavonoids have been extensively studied in various plant species, our understanding of these processes and their role in tuber color formation in *D. alata* remains limited.

Metabolite profiling and transcriptomic analyses have proven to be powerful tools for unraveling the biosynthesis of secondary metabolites in plants. Recent studies employing these techniques have shed light on the mechanisms underlying fruit color development in various plant species. For instance, a recent study demonstrated that two cultivars of plums with distinct fruit colors exhibit significant differences in gene expression related to flavonoid biosynthesis [29]. This disparity in gene expression contributes to variations in anthocyanin production and subsequent color changes in plum fruit. Similarly, another investigation employed integrated transcriptome and metabolome analyses to identify differentially expressed genes (DEGs) and metabolites (DEMs) in *Radix Ardisia* from different origins [30]. In that study, a total of 100 DEGs were identified related to flavonoid biosynthesis, including *4CL*, *AOMT*, *CHS*, *CHI*, *DFR*, *F3′5′H*, *FLS*, *LAR*, and others. Correlation analysis highlighted the potential key compounds in the flavonoid biosynthesis pathway, such as naringenin, luteolin, catechin, and quercetin [30]. In the same flavonoid synthesis pathway, differentially expressed genes and metabolites in different plants will also be different.

To clarify the mechanism of the yam tuber color formation, we performed metabolite profiling and transcriptomic analyses to investigate the differential expression of genes and metabolites in flavonoid biosynthesis in white and purple yam tubers in this study. Our findings will contribute to a better understanding of the flavonoid biosynthesis pathway in yam plants and provide insights into the differences in nutritional value and health benefits associated with white and purple yam tubers.

## 2. Results

### 2.1. Phenolic Compounds and Antioxidant Activities of White and Purple Tubers of D. alata

Six different *D. alata* cultivars were selected based on their tuber colors, namely ‘Suyu 6’ (Su6), ‘Ruianshanyao’ (RA), and ‘Minghuaishanyao’ (MH) and cultivars ‘S35’, ‘Gaotie’ (GT), and ‘Suyu 2’ (Su2). Cultivars Su6, RA, and MH exhibited white tubers (WTs), while cultivars ‘S35’, GT, and Su2 displayed purple tubers (PTs) (Figure 1A). The contents of the total phenols and total flavonoids in the WT and PT materials were determined by chemical analysis. The results showed that the PT samples exhibited relatively higher levels of total phenol and total flavonoid contents compared to the WTs. For example, the contents of the total phenols were 0.63, 0.76, and 0.70 mg/g in Su6, RA, and MH but reached 1.54, 2.48, and 2.61 mg/g in S35, GT, and Su2, respectively (Figure 1B). Similarly, the contents of the total flavonoids were 0.048, 0.052, and 0.050 mg/g in Su6, RA, and MH but reached 0.068, 0.081, and 0.087 mg/g in S35, GT, and Su2, respectively (Figure 1C). These findings suggested that the tuber color of the six *D. alata* cultivars may have been affected by the difference in the total phenolic and total flavonoid contents.

The antioxidant activities of the six *D. alata* cultivars were evaluated using two different methods (Table 1). The DPPH-scavenging activities of Su6, RA, and MH were 5.60 ± 0.105, 8.75 ± 0.071, and 6.46 ± 0.007 μmol Trolox/g, whereas the values for S35, GT, and Su2 were 16.84 ± 0.353, 23.07 ± 0.561, and 23.42 ± 0.412 μmol Trolox/g, respectively. Similar differences were obtained using ABTS radical assays, which showed that the scavenging activities of Su6, RA, and MH were 3.38 ± 0.055, 3.94 ± 0.108, and 3.53 ± 0.013 μmol Trolox/g, and the values for S35, GT, and Su2 were 4.01 ± 0.097, 6.45 ± 0.156, and 6.58 ± 0.393 μmol Trolox/g, respectively. These data collectively indicated that the antioxidant activities of the PTs were higher than those of the WTs among the six *D. alata* cultivars.

Based on the assays of the phenolic contents and antioxidant activities of the white and purple tubers of *D. alata*, correlation coefficients between the antioxidant activity and total phenolic/flavonoid content were analyzed (Table 2). The results showed that the contents of phenolics and flavonoids were closely correlated in the white (0.885) and purple (0.951) tubers. The DPPH and ABTS indexes of the antioxidant activity were more highly correlated with the phenolic content and flavonoid content in PTs than in WTs, indicating that purple tubers have a higher antioxidant activity. Thus, the correlation between the phenolics and antioxidant activity in yams is reasonable and could help to explain the different qualities of WT and PT yams.

### 2.2. Transcriptome Analysis of White and Purple Tubers from Two Different D. alata Cultivars

To investigate the underlying mechanism contributing to the differences in the tuber colors in *D. alata*, transcriptome sequencing was conducted separately on one WT (‘Suyu 6’) and one PT (‘Suyu 2’) sample. A total of 7443 DEGs were detected between the WT and PT, with 3350 genes showing down-regulated expression patterns and 4093 genes showing up-regulated expression patterns (Appendix A; Figure 2A,B). To validate the reliability of the RNA-seq results, 7 selected DEGs, including *DaANS*, *DaCHI*, *DaCHS*, *DaCYP73A*, *DaF3′5′H*, *DaFLS*, and *DaUGT78D2*, were subjected to qRT-PCR analysis. The results showed that the selected genes have a similar trend in gene expression, as observed in the RNA-seq data, supporting the reliability of the RNA-seq results in this study (Figure 2C).

The DEGs were annotated using the Kyoto Encyclopedia of Genes and Genomes (KEGG) database, and a KEGG pathway analysis was performed. The analysis revealed the representation of 138 pathways in the transcriptome dataset (Appendix A). The top 20 enriched pathways are illustrated in Figure 3. The rich factor refers to the ratio between the sample number of differential genes enriched in this pathway and the background number of annotated genes: the greater the rich factor, the greater the degree of enrichment, and the lower the *q* value, the more significant the enrichment. The first three terms with significant enrichment of differentially expressed genes are observed in metabolic pathways, secondary metabolite biosynthesis, and flavonoid biosynthesis pathways.

### 2.3. DEGs Related to Flavonoid Biosynthesis

Based on the results of the KEGG enrichment analysis, several pathways potentially associated with tuber color differences were identified, including flavonoid biosynthesis (ko00941, 57 genes); anthocyanin biosynthesis (ko00942, 2 genes); isoflavonoid biosynthesis (ko00943, 30 genes); and flavone and flavonol biosynthesis (ko00944, 19 genes). These pathways contain a total of 108 DEGs in the WT and PT, including 82 up-regulated and 26 down-regulated genes (Appendix A). Within the ko00941 pathway, which is annotated as the flavonoid biosynthesis pathway, thirteen chalcone synthase (CHS), four chalcone isomerase (CHI), three dihydroflavonol-4-reductase (DFR), five flavonol synthase (FLS), one flavanone 3-hydroxylase (F3H), one flavonoid 3′,5′-hydroxylase (F3′5′H), one leucoanthocyanidin reductase (LAR), one anthocyanidin synthase (ANS), and one anthocyanidin reductase (ANR) encoding genes were found to be highly expressed in the PT samples. Among them, the expression level of *DaFLS* was the most significantly different between the PT and WT (Figure 2C). Additionally, the expression levels of two genes (Dioal.19G068700.v2.1 and Dioal.16G051500.v2.1) involved in anthocyanin biosynthesis (ko00942) in the PT were higher than those in the WT.

### 2.4. Metabolomics Analysis between the WT and PT

To investigate whether the observed differential gene expression contributed to the divergence in the metabolite biosynthesis and tuber color between the WT and PT, the metabolite composition and content of the WT and PT samples were detected using the UPLC–MS/MS method. A total of 424 metabolites were found to be differentially accumulated in the WT and PT tubers, with 83 metabolites being down-regulated and 341 metabolites being up-regulated (in the PT or WT) (Figure 4A; Appendix A). These DEMs were found to be associated with 85 KEGG pathways. The top 20 metabolic pathways include the biosynthesis of secondary metabolites, flavonoid biosynthesis, phenylpropanoid biosynthesis, and flavone and flavonol biosynthesis (Figure 4B), which are well aligned with the results of the differential gene KEGG enrichment analysis.

### 2.5. Flavonoid and Tannin Metabolites in WT and PT

Based on the cluster analysis of the DEMs, it was found that flavonoids constituted the largest group, with a total of 104 DEMs belonging this group. These flavonoid DEMs include 3 anthocyanidins, 4 flavonoid carbonosides, 6 flavanols, 7 chalcones, 7 flavanonols, 17 flavanones, 18 flavones, and 42 flavonols. In addition to flavonoids, 8 tannins also showed significant differences between the WT and PT, mainly comprising proanthocyanidins (Figure 5A; Appendix A). Among the above 112 DEMs, most compounds showed an up-regulated pattern in the PT compared with the WT, except for two flavanones, hesperetin-7-*O*-neohesperidoside and hesperetin-7-*O*-rutinoside; they showed a down-regulated pattern in the PT compared with the WT. Notably, five of the top ten up-regulated metabolites were flavonoids, including 6-hydroxykaempferol-7-*O*-glucoside, pinocembrin-7-*O*-(6″-*O*-malonyl)glucoside, 6-hydroxykaempferol-3,7,6-*O*-triglycoside, 6-hydroxykaempferol-7-*O*-triglycoside, and cyanidin-3-*O*-(6″-*O*-feruloyl)sophoroside-5-*O*-glucoside, with the latter being a precursor to anthocyanin synthesis. Among the top 10 metabolic species with down-regulated levels, there were 2 flavonoids, namely hespertin-7-*O*-neohesperidoside (neohesperidin) and hespertin-7-*O*-rutinoside (hesperidin) (Figure 5B; Appendix A). These results indicate that the difference between the flavonoids and tannins in the PT and WT may be the main reason for the difference between the tuber colors.

### 2.6. Correlation Analysis between RNA-Seq and Metabolites Uncovers the Regulatory Pathway of Flavonoid Biosynthesis

To explore the mechanisms that underlie the difference between the tuber colors of the WT and PT, correlation analyses were performed between the gene expression and metabolite accumulation. The DEGs and DEMs of the same comparison group were mapped onto the KEGG pathway to find the co-enriched pathways. Based on the transcriptome results, the top 25 signaling pathways are shown (Figure 6A,B). Among the major KEGG pathways, we focused on the flavonoid biosynthesis (ko00941) and flavone and flavonol biosynthesis (ko00944) pathways, which may be related to the changes in the tuber color. In addition, all the DEGs and DEMs were selected to establish an O2PLS model. The variables with a high correlation and weight in different datasets were preliminarily determined through the loading maps, and important variables affecting other omics were screened out. The distance from each point to the origin or the height of the bar chart represents the magnitude of the correlation between a substance and another omic, and the darker the color indicates the greater the correlation. The top 10 substances that have a greater impact on other omics are indicated in the chart (Figure 6C,D). Among the top 10 genes and metabolites, the Dioal.09G026200.v2.1 and Dioal.19G168800.v2.1 genes belong to flavonoid biosynthesis, and HJAP135, Zmhp003514, pmb0580, and Hmcp002316 are flavonoids.

The flavonoid biosynthesis pathway showed the most significant correlation, with 57 differential genes and 20 differential metabolites associated with it (Figure 7; Appendix A). These metabolites included naringenin chalcone; 3,4,2′,4′,6′-pentahydroxychalcone; phlorizin chalcone; phloretin-2′-*O*-glucoside; 3′,4,4′,5,7-pentahydroxyflavan; epicatechin; epigallocatechin; naringenin; 5,7,3′,4′-tetrahydroxyflavanone; homoeriodictyol; hesperetin; naringenin-7-*O*-glucoside*; hesperetin-7-*O*-glucoside; neohesperidin; pinobanksin*; dihydrokaempferol; and dihydroquercetin. Compared with the WT, the PT contains a low content of neohesperidin but higher contents of the other metabolites. Notably, naringenin chalcone; 3,4,2′,4′,6′-pentahydroxychalcone; naringenin; 5,7,3′,4′-tetrahydroxyflavanone; homoeriodictyol; hesperetin; pinobanksin; and dihydrokaempferol were only detected in the PT. The associated genes included *CHS*, *CHI*, *F3H*, *DFR*, *ANS*, and *ANR*, and their expressions in the PT showed up-regulated patterns compared with those in the WT.

## 3. Discussion

### 3.1. D. alata Tubers Exhibiting Different Colors Possess Different Polyphenol Contents and Antioxidant Activities

*D. alata* is one of the most economically important crops in the tropical and subtropical regions; however, although it has a local significance, it is neglected on a global scale. Therefore, it may be considered as an orphan crop. Increasing the diversity of crops in global and local markets is one of the major challenges for agriculture [31]. Orphan crops offer greater nutritional value that could improve and diversify our diets [32]. *D. alata* may have potential as a functional food and could capture new market shares in the future. In this study, we investigated the differences between the polyphenol contents and antioxidant activities in *D. alata* tubers exhibiting different colors. Flavonoids, a significant group of polyphenols, are well-known for their influences on plant color changes [33,34] and their antioxidant properties [35]. To determine the polyphenol contents, we measured the levels of total phenols, total flavonoids, and proanthocyanidins in tubers with different colors. Our results demonstrated that white tubers displayed low levels of these compounds, whereas purple tubers exhibited high contents. This stark contrast in polyphenol contents offers a plausible explanation for the distinct coloration observed between the six *D. alata* cultivars.

Previous studies have consistently demonstrated a positive correlation between polyphenol and flavonoid contents and antioxidant activity [36]. Additionally, investigations of the polyphenol extracts from leaves of different yam varieties have confirmed their antioxidant activities [35]. Among the various flavonoids, anthocyanins, the most abundant compounds responsible for diverse colors [10,11,12], have been identified in yam tubers. Furthermore, it has been shown that purple yam tubers contain anthocyanins with distinct structures, resulting in varying levels of antioxidant activity [37]. However, previous studies lack a direct comparison of the antioxidant activities of polyphenol extracts from tubers of different colors. In our study, we measured the DPPH and ABTS indexes of polyphenol extracts from tubers with different colors and discovered that purple tubers exhibited the strongest antioxidant activity. Considering the well-established health benefits associated with antioxidant-rich foods, yam cultivars with purple tubers hold great promise for producing health-promoting foods and beverages.

### 3.2. Analysis of DEGs and DEMs Refers to Flavonoid Pathway in WT and PT of D. alata

Purple tuber yams have potential value as a health food, but the molecular mechanism of purple tuber formation is relatively unknown. A previous transcriptome analysis has revealed that several genes of the flavonoid pathway (including *CHS*, *F3H*, *F3′H*, *DFR*, *LDOX*, and UF3GT) were significantly up-regulated in purple yam tubers, implying that these genes were potentially associated with tuber color formation [1]. Although the changes in gene expression could affect the accumulation of metabolites [26,34,38] direct evidence of the metabolite profiles and their difference between purple and white yam tubers is still lacking. In this research, through transcriptome analysis, 108 differentially expressed genes of the flavonoid synthesis pathway between the WT and PT were identified, including CHS, CHI, DFR, FLS, F3′5′H, LAR, F3H, ANS, and ANR (Appendix A). Consistent with a previous study [1], we found that the expression levels of *CHS*, *F3H*, *F3*′*5*′*H*, *DFR*, and *LDOX* were higher in the PT than in the WT, but there was one gene, *FLS*, that was not mentioned in the previous study. In this study, compared with the WT, the PT possesses an *FLS* expression level that is increased 100-fold (Figure 2C). In other words, our results provide more comprehensive information to help researchers to understand the reason for the differences in tubers’ color formation.

To assess the impacts of DEGs on metabolite biosynthesis, we conducted metabolomics analysis and detected that 104 flavonoids and 8 tannins were differentially accumulated in purple and white tubers (Figure 5A; Appendix A). Previous studies have reported that *D. alata* can produce cyanidin-based anthocyanins, such as cyanidin 3-*O*-gentiobioside, alatanin 1, and alatanin 2 [3,39]. In support of this, three anthocyanins were found to be differentially accumulated between the WT and PT in this study, including alatanin 2, cyanidin-3-*O*-(6″-*O*-feruloyl) sophoroside-5-*O*-glucoside, and peonidin-3-*O*-(6″-*O*-feruloyl) sophoroside-5-*O*-glucoside. Although the content of alatanin 2 in the PT was significantly higher than that in the WT, the latter two were specifically accumulated in the PT. Moreover, 30 key metabolites related to the flavonoid synthesis pathway, including naringenin chalcone and naringenin, were specifically accumulated in the PT (Appendix A). Naringenin chalcone is the precursor of naringenin; they are upstream of the flavonoid synthesis pathway [40]. It is suggested that the formation of white tubers’ color may be caused by the synthesis of precursor substances.

To gain deeper insights into the molecular mechanism of pigment accumulation in tubers, we combined the transcriptome and metabolomics data. The combined analysis revealed the schema of the gene expression and metabolite accumulation in the flavonoid synthesis pathway of *D. alata* (Figure 7). Notably, CHS was found to regulate the accumulation of naringenin chalcone, while CHI catalyzed naringenin chalcone to generate naringenin, a crucial precursor of anthocyanin synthesis [17,27]. Under the actions of F3H, F3′5′H, and other up-regulated enzymes, DHK and DHQ were formed, serving as important precursors of anthocyanin. The expression levels of these genes and metabolites were significantly higher in the PT compared with the WT. Previous research has shown that anthocyanidin reductase (ANR) converts anthocyanidins, such as pelargonidin, cyanidin, and delphinidin, to corresponding *cis*-flavan-3-ols, including epiafzelechin, epicatechin, and epigallocatechin [41]. These results indirectly suggest that the content of anthocyanidins in the PT was higher than that in the WT, further supporting the pivotal role of the flavonoid synthesis pathway in regulating pigment accumulation in *D. alata* tubers.

## 4. Materials and Methods

### 4.1. Plant Materials

Six cultivars of yam plants (*D. alata*), including three cultivars of white tubers (‘Suyu 6’, ‘Ruian’, and ‘Minghuai’) and three cultivars of purple tubers (‘S35’, ‘Gaotie’, and ‘Suyu 2′), were used in this study. The yam plants were planted in a germplasm nursery at the Institute of Botany, Chinese Academy of Sciences, Jiangsu Province, and were cultivated using conventional irrigation and fertilization methods under natural conditions. The soil is sandy; the sand content is about 50%. The tubers were collected at the maturation stage for further analysis. Three biological replicates were set for each sample. For molecular analysis, the tuber samples were immediately snap-frozen in liquid nitrogen and stored at −80 °C until further use.

### 4.2. Total Polyphenol Extraction and Contents and Antioxidant Activity Analysis

To extract the total polyphenols, 100 mg of the sample powder was dissolved in 1.2 mL of a 70% methanol extraction solution. The sample was vortexed for 30 s every 30 min for 6 times. The sample was then stored overnight at 4 °C. Subsequently, the sample was centrifuged at 12,000 rpm for 10 min, and the supernatant was collected. The sample was filtered through a 0.22 μm pore-size membrane and stored in a sample vial for further analysis.

The total content of the soluble polyphenols was determined using the Folin–Ciocalteu method [42], with gallic acid used for standard calibration. The total content of the soluble flavonoids was measured according to [43], with rutin used for standard calibration. The antioxidant activity of the polyphenol extracts was measured using the 2,2-diphenyl-1-picrylhydrazyl (DPPH) and 2,2′-azinobis-(3-ethylbenzothiazoline-6-sulfonic acid) (ABTS) methods [44,45].

### 4.3. RNA Extraction, Library Preparation, and Sequencing

The total RNA was extracted from the samples using the TRIzol^TM^ Reagent, and the concentration of the total RNA was measured using a Quit 2.0 fluorescence spectrophotometer (Thermo, Waltham, MA, USA). The integrity of the total RNA was assessed using an Agilent 2100 bioanalyzer (Agilent Technologies, Palo Alto, CA, USA). Ribosomal RNA was removed from the total RNA, and the resulting RNA was fragmented. The fragmented RNA was then used as a template for first-strand cDNA synthesis, which was then used as templates to generate double-stranded DNA. The purified double-stranded DNA was subjected to end repair, A-tailing, sequencing adaptor ligation, and size-selection using AMPure XP beads to obtain the final cDNA library through PCR enrichment. Sequencing was performed using the Illumina HiSeq platform produced by the Wuhan Metware Biotechnology Co., Ltd. (Wuhan, China).

### 4.4. Transcriptome Data Analysis

The raw data of the RNA-seq was processed using Fastp to obtain clean reads for subsequent analysis [46]. The clean reads were then aligned to the reference genome of *D. alata* [47] using HISAT2 [48]. Then, StringTie was used to assemble the reads mapped to the reference genome into transcripts based on the location information in the genome. FPKM (fragments per kilobase of transcript per million fragments mapped) was used to measure the transcript or gene expression levels. The differential expression analysis between the sample groups was conducted using DESeq2 [49]. The screening criteria for the differentially expressed genes were |log2Fold Change| ≥ 1 and FDR (false discovery rate) < 0.05. The differentially expressed genes were annotated and enriched with KEGG, GO, and KOG.

### 4.5. Real-Time Quantitative Reverse-Transcription PCR Analysis

First-strand cDNA was synthesized using the total RNA that was extracted from the yam tuber samples. Real-time quantitative PCR (qPCR) was performed to validate the expression levels of the differentially expressed genes involved in flavonoid biosynthesis. The relative expression level of each gene was calculated using the 2^−ΔΔCT^ method [50], with ACTIN used as the reference gene. The analysis included three biological replicates for each cultivar, and three technical replicates were performed for each sample. The primer sequences used for qPCR are provided in Appendix A.

### 4.6. UPLC–MS/MS Conditions

Liquid-phase analysis was conducted using an ultra-performance liquid chromatography system (SHIMADZU Nexera X2, Kyoto, Japan) equipped with an Agilent SB-C18 1.8 µm × 2.1 mm × 100 mm column. The mobile phase consisted of A: ultrapure water (with 0.1% formic acid) and B: acetonitrile (with 0.1% formic acid). The flow rate was set at 0.35 mL/min, and the column temperature was maintained at 40 °C. The sample injection volume was 4 µL, and the elution gradient proceeded as follows: starting with 5% of the B phase, linearly increasing to 95% of the B phase within 9.00 min, holding at 95% for 1 min, reducing to 5% at 10.00–11.10 min, and equilibrating at 5% until 14 min.

The mass spectrometry analysis was performed using an Applied Biosystems 4500 Q TRAP tandem mass spectrometer (Applied Biosystems Inc., Foster City, CA, USA) equipped with an ESI Turbo ion spray interface. The instrument was operated in both the positive and negative ion modes. The ESI source operating parameters include a turbo spray source, a source temperature of 550 °C, and ion spray voltages (ISs) of 5500 V (in the positive ion mode)/−4500 V (in the negative ion mode). The gas settings of I (GSI), II (GSII), and the curtain gas (CUR) were 50, 60, and 25.0 psi, respectively. The collision-induced dissociation (CID) parameters were set to high. The instrument tuning and mass calibration were performed using 10 and 100 μmol/L polypropylene glycol solutions in the QQQ and LIT modes, respectively. QQQ scans were performed in the MRM mode with the collision gas (nitrogen) set to medium. The DP and CE for each MRM ion pair were optimized based on the elution of the metabolites in each period. A specific MRM ion pair was monitored during each period.

### 4.7. Differential Metabolite Analysis

To identify candidate differentially expressed metabolites (DEMs), a combination of univariate statistical analysis using parametric and non-parametric tests and multivariate statistical analysis methods, such as principal component analysis (PCA) and orthogonal projection to latent structure-discriminant analysis (OPLS-DA), was employed. The variable importance of projection (VIP) score from the application (O) PLS model was used to filter the metabolites showing the greatest differentiation between the groups. The parameters used for screening the DEMs were set at VIP ≥ 1 and fold change ≥ 2.0 or ≤0.50. Subsequently, the identified DEMs were mapped to KEGG metabolic pathways for pathway and enrichment analyses at a significance threshold of a *p*-value of <0.05.

### 4.8. Combined Transcriptome and Metabolome Analyses

The metabolite and transcriptomic data were analyzed using multivariate statistical techniques to identify significant differences in metabolite and gene expressions between the white and purple yam tubers. The KEGG enrichment analysis of the differentially expressed metabolites and genes was performed to identify commonly enriched pathways in both groups. The ‘cor’ function in R was utilized to calculate the Pearson correlation coefficient between the genes and metabolites. The correlation results with a correlation coefficient greater than 0.80 and a *p*-value of less than 0.05 were considered statistically significant. Canonical correlation analysis (CCA) was conducted on the differentially expressed genes and metabolites within each pathway. An O2PLS model was built using all the differentially expressed genes and metabolites. The loading plot of the model was examined to preliminarily identify variables with high correlations and weights in different data groups, enabling the screening of important variables that influence other groups of data.

## 5. Conclusions

In conclusion, our study revealed significant differences between the polyphenol contents and antioxidant activities in purple and white tubers of *D. alata*. The integrated analysis of the transcriptome and metabolomics data demonstrated that 57 genes, including *CHS*, *CHI*, *DFR*, *FLS*, *F3H*, *F3*′*5*′*H*, *LAR*, *ANS*, and *ANR*, played a regulatory role in altering the contents of 20 metabolites within the flavonoid synthesis pathway. Metabolites related to the flavonoid synthesis pathway, including naringenin chalcone and naringenin, were specifically accumulated in the PT. These genes and metabolites are likely to play a crucial role in the color formation of *D. alata* tubers. These findings provide valuable insights into the molecular basis of tuber color variation in *D. alata*, facilitating the development of purple tubers as nutritious and health-enhancing food options.

## Figures and Tables

**Figure 1 ijms-25-02057-f001:**
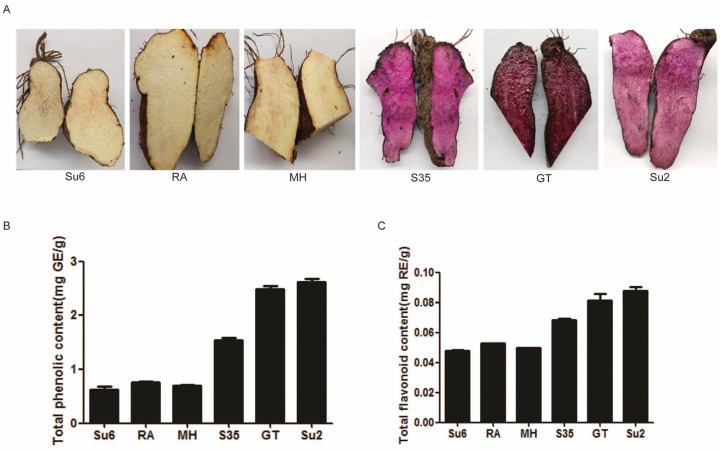
Tuber performance of different cultivars of yam plants. (**A**) The tuber colors of six cultivars; (**B**) total phenolic contents; (**C**) total flavonoid contents; Su6 represents ‘Suyu 6’ (white tuber); RA represents ‘Ruianshanyao’ (white tuber); MH represents ‘Minghuaishanyao’ (white tuber); GT represents ‘Gaotie’ (purple tuber); Su2 represents ‘Suyu 2’ (purple tuber). GE, gallic acid equivalents; RE, rutin equivalents.

**Figure 2 ijms-25-02057-f002:**
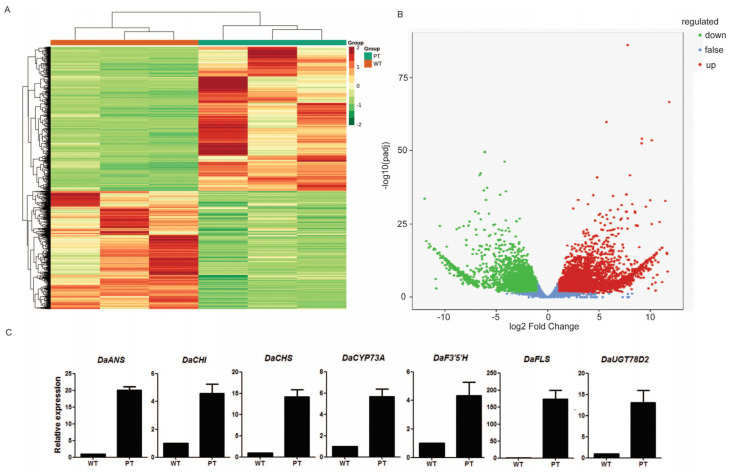
Differentially expressed genes between WT and PT, and qRT-PCR analysis of the results of RNA-seq. (**A**) Cluster heatmap of differentially expressed genes. (**B**) Volcano plot of differentially expressed genes. (**C**) Differentially expressed genes were confirmed by qRT-PCR. WT: white tuber; PT: purple tuber. Data are represented as mean values ± SD, *n* = 3.

**Figure 3 ijms-25-02057-f003:**
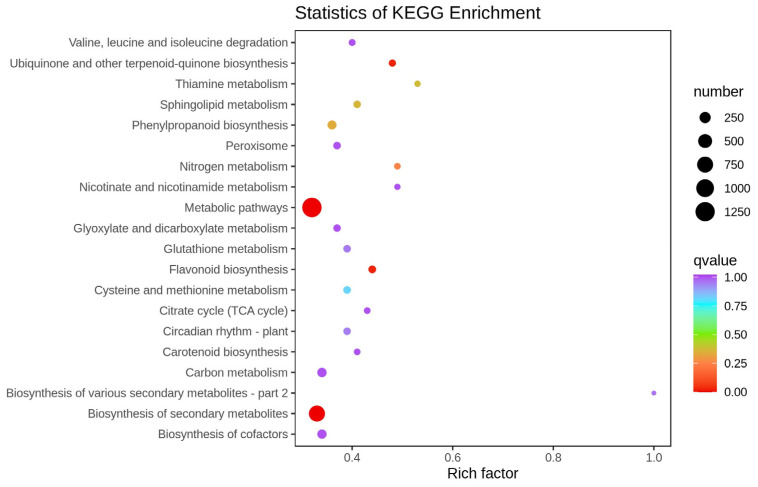
Top 20 terms of KEGG enrichment for DEGs between PT and WT.

**Figure 4 ijms-25-02057-f004:**
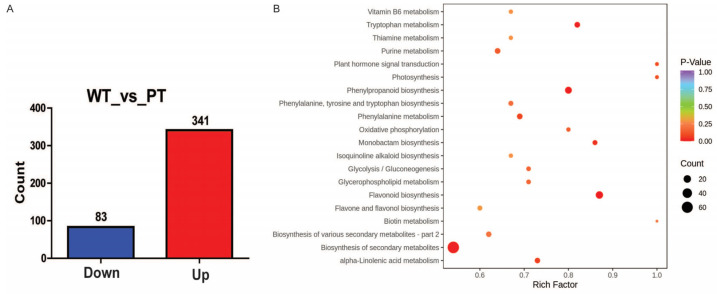
Differential metabolite analysis between WT and PT. (**A**) The number of differentially expressed metabolites. (**B**) Scatter plot of top 20 enriched pathways of differentially expressed metabolites. WT: white tuber; PT: purple tuber.

**Figure 5 ijms-25-02057-f005:**
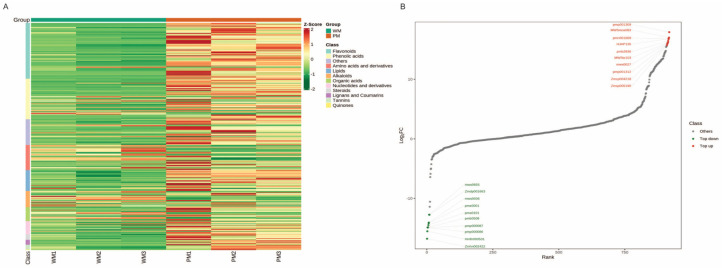
Analysis of differentially expressed metabolites between the WT and PT. (**A**) Heatmap of metabolite contents (by substance). (**B**) Dynamic distribution of metabolite content differences. The horizontal coordinate represents the cumulative number of substances arranged in the order of the difference multiple from low to high; each point represents a substance: the green points represent substances that are down-regulated in the top 10, and the red points represent substances that are up-regulated in the top 10.

**Figure 6 ijms-25-02057-f006:**
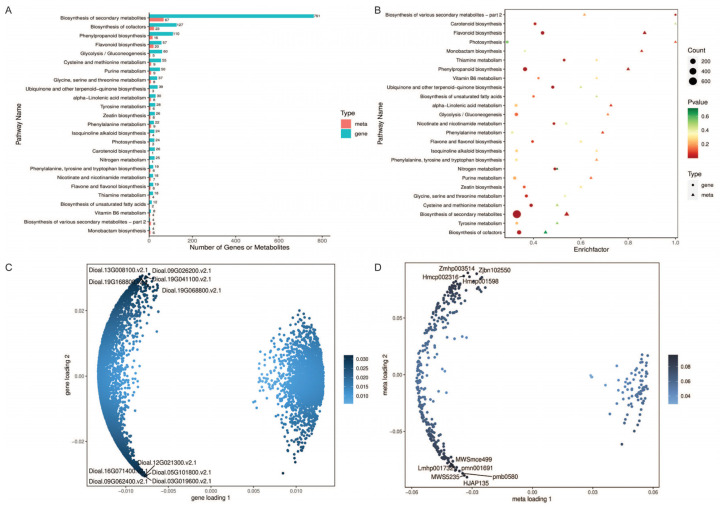
Co-enrichment analysis of DEGs and DEMs. (**A**) Bar chart of KEGG enrichment analysis; (**B**) bubble map of KEGG enrichment analysis; (**C**) DEG loading map; (**D**) DEM loading map.

**Figure 7 ijms-25-02057-f007:**
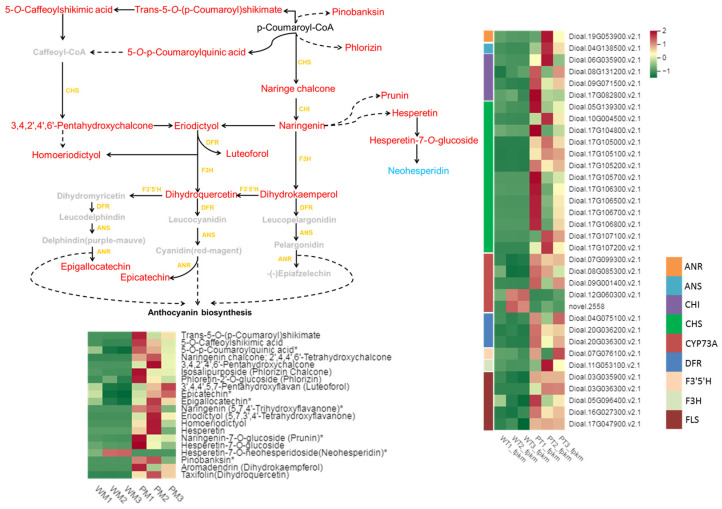
Schematic of flavonoid-related genes and metabolites in WT and PT. Red fonts indicate increased metabolite content, blue fonts indicate decreased metabolite content, gray fonts indicate metabolites with no change in content, and yellow fonts indicate genes that influence changes in metabolite content. * indicates the presence of isomers.

**Table 1 ijms-25-02057-t001:** Antioxidant activities of different colored tubers, as determined using DPPH and ABTS assays.

Sample	DPPH	ABTS
No.	(μmol Trolox/g DW ± SE)	(μmol Trolox/g DW ± SE)
Su6	5.60 ± 0.105	3.69 ±0.092
RA	8.75 ± 0.071	3.95 ± 0.108
MH	6.46 ± 0.007	3.53 ± 0.013
S35	16.84 ± 0.353	4.01 ± 0.097
GT	25.85 ± 2.222	6.45 ± 0.156
Su2	23.06 ± 0.662	6.58 ± 0.393

Note: Data represent the mean values ± standard error (*n* = 3).

**Table 2 ijms-25-02057-t002:** Correlation coefficient analysis of phenolic compounds and antioxidant activities between WTs and PTs.

	Flavonoid		ABTS		DPPH	
	WT	PT	WT	PT	WT	PT
Phenolics	0.885 *	0.951 **	0.887 *	0.945 **	0.782	0.980 **
Flavonoids	-	-	0.656	0.835 **	0.931 **	0.917 **

Note: “*” and “**” indicate significance at *p* < 0.05 and 0.001 levels, respectively.

## Data Availability

Data are contained within the article and Appendix A.

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
