# Peer review of "Unraveling the Molecular Basis of Color Variation in Dioscorea alata Tubers: Integrated Transcriptome and Metabolomics Analysis"

_ijms, 2024, doi:10.3390/ijms25042057_

Round 1

Reviewer 1 Report

Comments and Suggestions for Authors

Dear Sir/Madam,

Thank you for your effort. The manuscript entitled: 'Unraveling the Molecular Basis of Color Variation in Dioscorea alata Tu- 2 bers: Integrated Transcriptome and Metabolomics Analysis'  is very well written, with an appropriate design and compares differentially expressed genes in white and purple yam tubers. Below the authors can find the minor changes to the manuscript:

Line 35: replace with: root tuber belonging to the Dioscoreaceae Family

Line 36: replace with: various regions

Line 37: replace with: exhibits a range of colors

Line 39: replace with: with color variation generally attributed to

Line 59: replace with: that catalyzes the reduction of DHK

Line 72: In the study, a total number of 100 DEGs have been identified related to flavonoid biosynthesis

Line 83: Six different D. alata cultivars have been selected based on their tuber colors, namely:

Line 201: To explore them mechanisms that underline the differences in tube color between WT and PT, correlation analyses have been performed...

Comments on the Quality of English Language

The English language is fine. Minor changes in the manuscript need to be made

Reviewer 2 Report

Comments and Suggestions for Authors

Dear authors!

Thank you for submitting your work.

The article is devoted to the study of the content of phenols and flavonoids in the tissues of six different varieties of the cultivated plant Dioscorea alata. In general, the main idea of the article does not seem new to me. It has long been known that the intensity of the color of plant fruits is associated with the content of phenolic substances and flavonoids. The only novelty in this article is the object on which the research was conducted. However, the worldwide distribution and intensity of cultivation of this plant is limited, which narrows the interest in the article.

There are sections in the article in which the information is formatted absolutely not according to the rules of the "IJMS"; this characterizes the careless attitude of the authors of the article towards the editors and reviewers of the journal (for example, some words do not correspond to the required font size, quotation marks are not formatted according to the rules, sometimes in paragraphs the text is written in bold, there are negligence in the design of the list of references). Authors must format the text in accordance with the journal's rules.

The main problem with the above article is this conclusion “study revealed significant differences in polyphenol content and antioxidant activity between purple and other color tubers.” There is no novelty in it. If you want to successfully publish your manuscript, then you need to focus attention in the conclusion, abstract, and overall article on your other results, for example, those obtained using transcriptomic and metabolomic analyses.

It is also recommended that in the “Materials and Methods” paragraph it is advisable to indicate the composition and characteristics of the soil on which the plants were grown.

Drawings often contain completely unreadable captions; this needs to be corrected. Photos in Fig. 1 are unsuccessful, because contain cropped images of tubers.

Unfortunately, the article in its present form cannot be recommended for publication.

Respectfully Yours, reviewer

January 16, 2024

Round 2

Reviewer 2 Report

Comments and Suggestions for Authors

Dear authors, thank you for the work you have done to change the article.

Now the emphasis of the article has been rearranged to the necessary points.

However, the literary sources that you added to the article, in my opinion, are outdated. More recent links could have been found.

I leave the decision on whether to accept the article to the editor.

Respectfully Yours, reviewer

January 26, 2024
